# The Role of Perceived Justice on Satisfaction with the Coach: Gender Differences in a Longitudinal Study

Miguel Ángel López-Gajardo 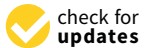, José Carlos Ponce-Bordón, Ana Rubio-Morales, Rubén Llanos-Muñoz and Jesús Díaz-García *

Faculty of Sport Sciences, University of Extremadura, 10003 Cáceres, Spain; malopezgajardo@unex.es (M.Á.L.-G.); jponcebo@gmail.com (J.C.P.-B.); anarubmor94@gmail.com (A.R.-M.); rubenllanosm98@gmail.com (R.L.-M.)
* Correspondence: jdiaz@unex.es

**Abstract:** The aim of this study was to determine the interaction between the factors of perceived justice with regard to players' satisfaction with the coach's behavior, and also to determine the evolution of these interactions across a season in elite male and female soccer. A longitudinal design was carried out, with three measurement points (i.e., at the beginning, in the middle, and at the end of a season). Participants were 439 professional soccer players (males = 227, females = 212), aged between 18 and 33 years ($M$ = 23.81, $SD$ = 4.53). Results showed gender differences in the factors that predict satisfaction with the coach. Women grant more importance to relational and motivational aspects. It was also confirmed that there are important variations across the season in both genders. These results can help to better understand which behaviors are more appropriate for coaches depending on gender and time of the season.

**Keywords:** coach behavior; fairness; player perceptions; professional players; gender

## 1. Introduction

Sports training, and specifically soccer, has evolved until the appearance of new formulas that seek to collect the different capacities necessary to achieve the adaptation of the player to their contextual reality and in order to optimize sports performance. These capacities include not only physical aspects, but also psychological aspects such as emotions, motivations, cognitive and socio-affective aspects, etc. The work of the coach in all of them is key to maintaining or improving the performance of the players. Along these lines, one of sports coaches' main concerns is for their athletes to be satisfied with coach behavior, and for them to consider that the coach is adequately performing his/her professional and interpersonal function [1]. This fact has led to an interesting line of research to discover which variables can best determine players' high satisfaction with their coach's work, and with the behavioral benefits derived thereof, such as higher levels of commitment, effort, or performance [2–4]. Among the variables that have been used to predict players' satisfaction with coaching performance, their perception of how coaches impart justice with their decisions could be particularly interesting, although we do not know previous studies that have verified this question. Thus, this investigation will deepen the knowledge of how players' perception of coaching justice can predict their satisfaction with the coach's work. This analysis will be carried out with a longitudinal design, as the variables are dynamic and fluctuate across the season and, to date, no study has analyzed this evolution. We also intended to determine whether there are gender differences in the perception of these constructs and in their interrelationship across the season in a professional sports context.

### 1.1. Coach Behavior in Sport

The analysis of coaches' perceived behavior and its relation to behavioral variables has generated many studies in recent years from different conceptual perspectives. Many

of these theoretical approaches to the study of coach behavior have focused on analyzing the leadership model of the coach and its relation to other variables [5], the motivational climate in the training sessions [2,4,6], or the effectiveness of identifying the characteristics needed to optimize team functioning or to achieve team well-being [7].

In spite of this, these theoretical frameworks have difficulties in explaining certain behaviors in the context of professional sport, and a more specific and complex analysis interlinking different constructs is needed [8,9]. Therefore, in the sport context, other theoretical frameworks have recently emerged to explain the relations between coaches and players in the professional context, adapted from the psychology of organizations due to its similarity to the sports context. In this sense, perceived coaching justice emerges as an interesting construct, which has been used to try to expand our knowledge of the figure of the coach and its relation with group satisfaction and well-being [10,11].

### 1.2. Perceived Justice

From the perspective of psychology of organizations, perceived justice was found to be an essential element to generate adequate satisfaction in the work environment and with the supervisor's work [12–14]. Along this line, in a professional sport context, perceived coach justice can be considered a relevant aspect to promote an adequate work environment and to improve satisfaction and performance [11,15,16]. If the players perceive that the coach's decisions when interacting with the players and in the procedures used to distribute rewards are just, then their satisfaction with his/her work and the group environment will be reinforced [10,16–19].

Perceived justice was conceptualized by Greenberg [20] as "grown around attempts to describe and explain the role of fairness as a consideration in the workplace" (p. 400). There are many classifications of the dimensions of perceived justice, but following the tenets of Colquitt [21], we conceive this construct as having four factors. First, research focused on distributive justice (DJ), which describes the fairness of an employee's outcomes, especially the degree to which they are equitable. Playing time and assigned position or role are typical examples of outcomes in team sports [11]. Second, procedural justice (PJ) reflects the perceived fairness of decision-making processes and the degree to which they are consistent, accurate, and ethical [21]. In a sport context, an athlete may be dissatisfied with the selected team captain. However, if the procedures used to select the captain are perceived as fair, the athlete is more likely to accept the final decision [11]. Third, informational justice (INFJ) is the perceived adequacy of explanations of decision processes and outcomes, and of why procedures were used in a certain way or how outcomes were distributed [12]. In team sports, explaining the criteria that will be used to select starting players or replacements during the match can increase the likelihood that all team members will accept the final decisions as fair [11]. Fourth, interpersonal justice (INTJ) is the perceived degree of dignity and respect shown by the authorities—coach— who are involved in the procedures to achieve outcomes.

Regarding the relation between perceived justice and satisfaction with head coach (SHC), to our knowledge, no work has related these variables, although there are some studies with similar constructs. Thus, Nikbin et al. [19] found that players' satisfaction with their role and participation was mainly related to interactional justice (INTJ and INFJ), although DJ and PJ also had positive effects. Similarly, De Backer et al. [10,17] related justice positively to need for support from the coach, mastery climate, task cohesion, and identification with the team, whereas they were negatively related to performance climate and social loafing. Lastly, various works [11,15,19], have confirmed the importance of perceived justice on team unity, cohesion, and commitment to the team, variables that could have an impact on the final satisfaction with the coach.

### 1.3. The Present Study

The goal of this study was to determine the impact of perceived coach justice on soccer players' satisfaction with their coach's management of the team, trying to appraise the

changes that occur over a season in these relationships, and analyzing differences between male and female elite teams.

In this way, our study extends past literature by employing a longitudinal design (season-long) to examine how the variables evolve over the season and the degree to which players´ perceptions of coaching behavior of justice are related to their SHC. By examining these associations across a sport season, we tried to shed light on the dynamics of these associations because we presumed that perceived justice and players' SHC fluctuates across time. Finally, we tried to analyze gender differences in perceived justice in their relationship with SHC, as well as differences in their evolution throughout the season. In order to meet these goals, we proposed a series of hypotheses to guide our investigation.

Thus, in spite of the fact that there are no previous studies to help us determine the evolution that perception of justice and SHC will undergo over the season, and that many variables might modify the perception of these factors, we hypothesized that the variables of the study would suffer a general decrease as the season progresses (Hypothesis 1). The reason for this is that, in previous longitudinal works with soccer teams, the variables with a positive connotation (learning climate, cohesion, role satisfaction, collective efficacy . . . ) decreased significantly and generally, although there were always some factors that reduced or augmented the decrease [4,8,22,23].

Second, we hypothesized that the factors of perceived justice would positively and significantly predict SHC in the three measures taken across the season (Hypothesis 2), as found in some prior works that have linked justice to different types of satisfaction [10,19,20]. However, we thought that these relations may vary in intensity although, as there are no studies that have analyzed the evolution of the interaction of these variables longitudinally, we did not dare to formulate a hypothesis in this regard because we did not know how the relationship with SHC of the factors of justice may vary as the season progresses.

Finally, we hypothesized that male players would grant more importance to factors of performance and task, such as PJ or DJ. On the other hand, the more support the coach grants to social and relational aspects, such as INFJ or INTJ, the more satisfied would female players feel with the coach (Hypothesis 3).

To support this hypothesis, we took into account the tenets of different theories and previous empirical studies in areas related to this work. Thus, based on the social role theory, women are typically described as, and expected to be, more relations-oriented and nurturing than men, whereas men are believed and expected to be more task-oriented and agentic than women [24,25]. This is also based on previous studies indicating that women grant more importance to social aspects and they have higher levels of empathy regarding an appropriate relationship with the coach, and they positively value leadership more focused on relations [1,26]. In contrast, males are more performance-oriented and prefer to have more freedom to make decisions and participate in the teaching–learning process [17,18].

## 2. Materials and Methods

### 2.1. Participants

Participants were professional male and female soccer players belonging to 31 professional teams. Specifically, 13 out of 15 female teams of Women's First National Division and 18 out of 20 male teams of Men's Third (Second B in Spain) National Division agreed to participate. In order to avoid possible errors due to the changes of players and coaches that take place throughout the season, we decided to eliminate from the sample the teams that had changed their coach, as well as the players who had not been on the team from the beginning to the end of the season. The final sample was therefore made up of 439 soccer players. Of these, 227 were males aged 18 to 33 years (*M* = 25.27, *SD* = 4.52), belonging to the 11 teams that kept their coach through the entire season, and 212 females aged 18 to 31 years (*M* = 23.81, *SD* = 4.53), of the 12 teams that kept their coach through the entire season.

## 2.2. Measures

Perceived Justice. To assess perceived justice, an adapted and translated Spanish version of Colquitt's Justice Questionnaire [2] was used. The scale has 12 items grouped into four factors: Procedural Justice, Distributive Justice, Interpersonal Justice, and Informational Justice. Players respond to all items on a seven-point scale ranging from never (1) to always (7). The confirmatory factor analysis (CFA) showed an adequate fit of the model: $\chi^2(166) = 233.57$, $p < 0.004$, CFI = 0.92, TLI = 0.91, SRMR = 0.06. Model fit was assessed using chi-square ($\chi^2$), comparative fit index (CFI), Tucker Lewis Index (TLI), and standardized root mean square residual (SRMR). CFI and TLI values equal to or greater than 0.90, and SRMR scores equal to or less than 0.06, respectively, were considered acceptable. In addition, this instrument showed acceptable levels of internal consistency ($\alpha_{\text{procedural justice}} = 0.81$, $\alpha_{\text{distributive justice}} = 0.82$, $\alpha_{\text{interpersonal justice}} = 0.79$, and $\alpha_{\text{informational justice}} = 0.83$).

Satisfaction with the Coach. To assess SHC, we used the adapted Spanish version [27] of the scale developed by Myers, Beauchamp, and Chase [28]. The scale contains three items. Responses were rated on a five-point scale ranging very little (1) to a lot (5). The CFA showed a strong fit of the model also in this sample: $\chi^2(5) = 35.38$, $p < 0.001$, CFI = 0.93, TLI = 0.93, SRMR = 0.05. Furthermore, this scale showed acceptable levels of internal consistency ($\alpha = 0.79$).

## 2.3. Procedure

We used a longitudinal correlational design. We carried out three assessments at three time points: at the beginning of the sport season (Time 0), in the middle (Time 1), and at the end of the season (Time 2), separated by a 20–22-week interval between each measurement wave. The study received ethical approval from the University. The principal investigator contacted the sport psychologists of the teams to explain the aims of the study and its confidential nature. Athletes' and coaching staff's consent was obtained. Therefore, all participants were treated according to the American Psychological Association's ethical guidelines regarding consent, confidentiality, and anonymity of responses. The procedure was similar at all timepoints (i.e., Time 0, 1, and 2). Participants completed the questionnaires in the changing room, individually within 15–20 min, in the absence of their coach, supervised by the research assistants and under non-distracting conditions.

## 2.4. Data Analysis

All analyses were performed using the software SPSS 25.0 (IBM SPSS Statistics for Windows, Version 25.0. IBM Corp, Armonk, NY, USA). Initially, reliability (Cronbach's alpha) and factorial validity (CFA) of all measures were analyzed, and the descriptive statistics were estimated for all study variables at the three time points. Analysis of Variance (ANOVA) was also used to examine the gender differences at each time point and between Time 0 and Time 2.

Secondly, we grouped the main analyses into two steps, developing different regression analyzes through mixed models and including SHC as the dependent variable in both cases. In the first step, we carried out two separate regression analyses (Model 1 with male players; Model 2 with female players) using the time point as a factor and the measure of perceived justice as covariate. We evaluated the main effects of these variables, as well as the interactions between time and predictors. In the second step, we carried out three separate regression analyses (Model 3 at Time 0; Model 4 at Time 1; Model 5 at Time 2) including the gender as a factor and perceived justice as covariate. We evaluated both the main effects and the interactions between gender and predictor.

## 3. Results

### 3.1. Preliminary Analysis

Means and standard deviations for all study variables by gender at the three time points are presented in Table 1. In terms of gender differences, males obtained significantly greater scores in PJ and DJ at Time 0; in PJ, DJ, INFJ, and INTJ at Time 1, and no differences

at Time 2. Regarding the evolution over the season, all variables had significantly lower scores at Time 2 than at Time 0. Men showed a large decrease in their values, while women's values decreased less, especially between Times 1 and 2.

**Table 1.** Means and standard deviations of all study variables at the three time points and difference by gender and between Time 0 and Time 2.

| | Time 0 (August) | | | Time 1 (December) | | | Time 2 (May) | | | Difference Time 0–2 |
|---|---|---|---|---|---|---|---|---|---|---|
| | **Male** | **Female** | $p$ | **Male** | **Female** | $p$ | **Male** | **Female** | $p$ | |
| PJ | 5.48 ± 0.92 | 5.15 ± 1.04 | <0.01 | 5.03 ± 1.16 | 4.35 ± 1.34 | <0.001 | 4.61 ± 1.30 | 4.46 ± 1.50 | 0.405 | <0.001 |
| DJ | 5.38 ± 1.08 | 5.00 ± 1.06 | <0.01 | 4.86 ± 1.27 | 4.10 ± 1.45 | <0.001 | 4.36 ± 1.48 | 4.16 ± 1.16 | 0.306 | <0.001 |
| INFJ | 6.24 ± 0.93 | 6.11 ± 0.81 | 0.288 | 5.92 ± 1.02 | 5.62 ± 0.29 | <0.01 | 5.47 ± 1.21 | 5.35 ± 1.29 | 0.489 | <0.001 |
| INTJ | 5.76 ± 0.86 | 5.63 ± 1.11 | 0.325 | 5.29 ± 1.25 | 4.96 ± 1.36 | <0.01 | 4.88 ± 1.39 | 4.90 ± 1.52 | 0.902 | <0.001 |
| SHC | 4.35 ± 0.66 | 4.34 ± 0.63 | 0.921 | 3.97 ± 1.00 | 3.98 ± 0.91 | 0.956 | 3.59 ± 1.13 | 3.68 ± 1.14 | 0.517 | <0.001 |

PJ = procedural justice; DJ = distributive justice; INFJ = informational justice; INTJ = interpersonal justice; SHC = satisfaction with the head coach.

*3.2. Main Analysis*

3.2.1. Interrelationship of Perceived Justice with SHC by Gender over a Competitive Season

Regarding the evolution of SHC over the season (Table 2), male players presented a significant decrease over the three time points (Time 1 vs. Time 0, $p < 0.05$; and Time 2 vs. Time 0, $p < 0.01$), whereas female players did not show significant changes in SHC at Time 1, but they did show significant changes at Time 2 ($p > 0.05$).

**Table 2.** Regression coefficients and standard errors of the evolution of the prediction of satisfaction with the coach by perceived justice for male and female players.

| | Model 1: Male Players | | | Model 2: Female Players | | |
|---|---|---|---|---|---|---|
| | $\beta$ | $T$ | $p$ | $\beta$ | $T$ | $p$ |
| Evolution of SHC over season | | | | | | |
| Intercept (Time 0) | 4.009 | 63.444 | <0.001 | 4.175 | 78.546 | <0.001 |
| Time 1 | −0.129 | −1.969 | 0.050 | −0.006 | −0.098 | 0.922 |
| Time 2 | −0.174 | −2.267 | 0.024 | −0.226 | −3.139 | 0.002 |
| Associations at Time 0 | | | | | | |
| PJ | 0.320 | 3.203 | 0.001 | 0.137 | 1.504 | 0.133 |
| DJ | 0.065 | 0.865 | 0.388 | −0.010 | −0.123 | 0.903 |
| INTJ | 0.132 | 1.884 | 0.060 | 0.183 | 2.507 | 0.013 |
| INFJ | 0.248 | 2.482 | 0.014 | 0.279 | 3.196 | 0.002 |
| Evolution of associations over season | | | | | | |
| Time 1 × PJ | −0.101 | −0.788 | 0.432 | 0.023 | 0.188 | 0.851 |
| Time 2 × PJ | −0.310 | −2.113 | 0.035 | 0.254 | 1.85 | 0.065 |
| Time 1 × DJ | 0.154 | 1.521 | 0.129 | 0.075 | 0.671 | 0.503 |
| Time 2 × DJ | 0.314 | 2.885 | 0.004 | 0.096 | 0.794 | 0.428 |
| Time 1 × INTJ | 0.039 | 0.448 | 0.654 | −0.041 | −0.474 | 0.636 |
| Time 2 × INTJ | −0.043 | −0.482 | 0.630 | −0.082 | −0.865 | 0.387 |
| Time 1 × INFJ | 0.077 | 0.641 | 0.522 | 0.094 | 0.835 | 0.405 |
| Time 2 × INFJ | 0.249 | 1.945 | 0.053 | 0.008 | 0.062 | 0.950 |

PJ = procedural justice; DJ = distributive justice; INFJ = informational justice; INTJ = interpersonal justice; SHC = satisfaction with the head coach.

Regarding the predictors of SHC, at Time 0, in the male group (Model 1), PJ ($\beta = 0.320$; $p = 0.001$) and INFJ ($\beta = 0.248$; $p = 0.014$) positively predicted SHC, whereas for female players (Model 2), INTJ ($\beta = 0.183$; $p = 0.013$) and INFJ ($\beta = 0.279$; $p = 0.002$) were positive and significant predictors of SHC.

In terms of evolution of the associations between predictors and the dependent variable, for male players, the association between PJ and SHC changed significantly over the season (Figure 1). Specifically, at Time 1, the degree of association was $\beta = 0.320 - 0.101 = 0.219$, whereas, at Time 2, the degree of association was $\beta = 0.320 - 0.322 = -0.002$ (Time 2 vs. Time 0, $p > 0.05$). Furthermore, the relation between DJ and SHC was significantly greater at Time 2 ($\beta = 0.065 + 0.314 = 0.379$, $p < 0.01$) compared to Time 0.

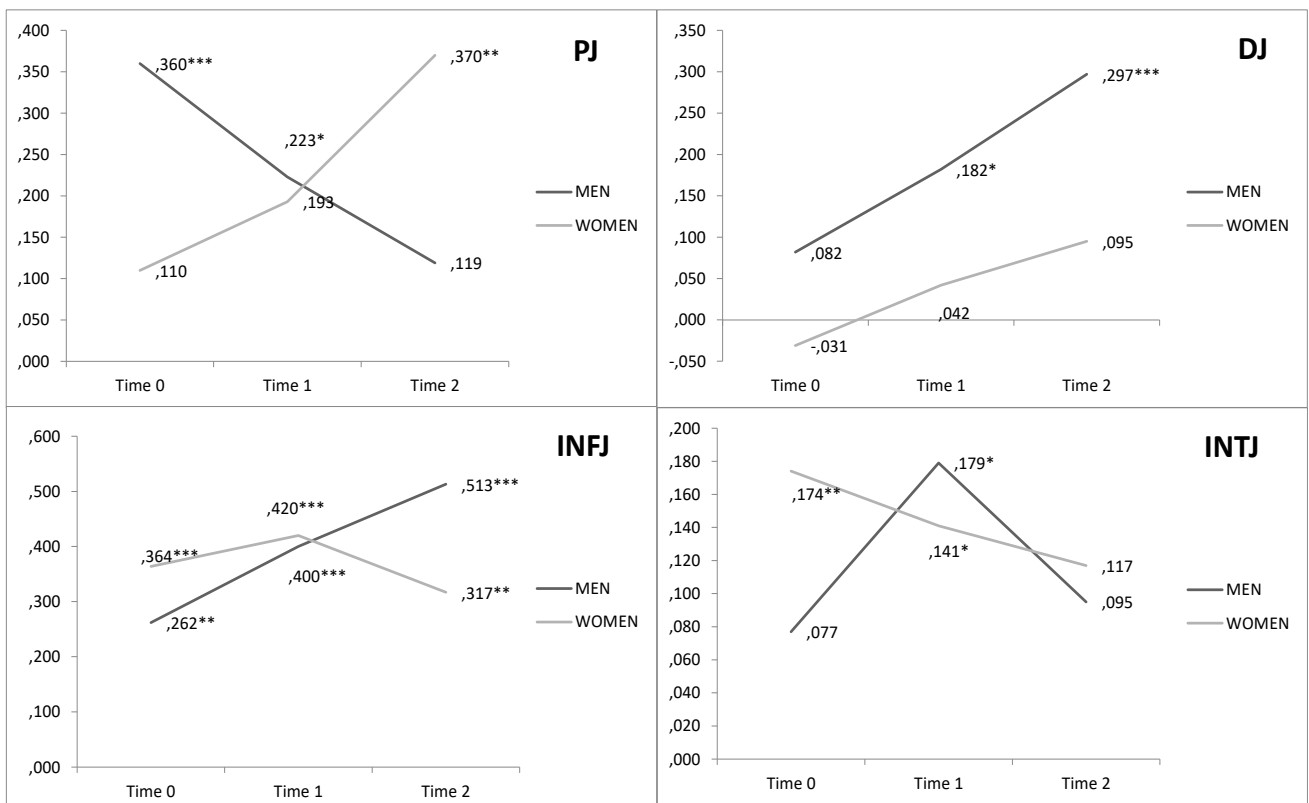

**Figure 1.** Graphic representation of the interrelationship between perceived justice and satisfaction with the coach for male and female players. PJ = procedural justice; DJ = distributive justice; INFJ = informational justice; INTJ = interpersonal justice; SHC = satisfaction with the head coach; * $p < 0.05$; ** $p < 0.01$; *** $p < 0.001$.

Regarding the female group, the positive and significant association of INTJ and INFJ with respect to SHC found at Time 0 remained over the three time points. However, the relation between INTJ and SHC descended significantly at Time 1 and 2, while the relation between PJ and SHC increased, although none of these changes became significant.

3.2.2. Gender Differences in the Associations between Perceived Justice and SHC at the Three Time Points

In order to examine the influence of gender on the interrelationship of perceived justice with SHC, we ran a model comparing both genders in the three moments of the season. Results are presented in Table 3.

**Table 3.** Regression coefficients and standard errors of the prediction of satisfaction with the coach by perceived justice at Time 0, Time 1, and Time 2 by gender.

| | Model 3: Time 0 | | | Model 4: Time 1 | | | Model 5: Time 2 | | |
|---|---|---|---|---|---|---|---|---|---|
| | $\beta$ | $T$ | $p$ | $\beta$ | $T$ | $p$ | $\beta$ | $T$ | $p$ |
| Gender differences | | | | | | | | | |
| Intercept (Males) | 3.999 | 75.551 | <0.001 | 3.880 | 72.966 | <0.001 | 3.844 | 68.781 | <0.001 |
| Females | 0.164 | 2.326 | 0.021 | 0.302 | 3.820 | <0.001 | 0.115 | 1.418 | 0.157 |
| Associations for male players | | | | | | | | | |
| PJ | 0.360 | 4.017 | <0.001 | 0.223 | 2.055 | 0.041 | 0.119 | 0.962 | 0.337 |
| DJ | 0.082 | 1.217 | 0.225 | 0.182 | 2.056 | 0.041 | 0.297 | 3.282 | <0.001 |
| INTJ | 0.067 | 1.084 | 0.279 | 0.179 | 2.321 | 0.021 | 0.095 | 1.449 | 0.149 |
| INFJ | 0.262 | 2.907 | 0.004 | 0.400 | 4.721 | <0.001 | 0.513 | 5.602 | <0.001 |
| Associations for female players | | | | | | | | | |
| PJ | 0.110 | 1.330 | 0.185 | 0.193 | 1.819 | 0.070 | 0.370 | 3.066 | 0.002 |
| DJ | −0.031 | −0.418 | 0.676 | 0.042 | 0.440 | 0.660 | 0.095 | 0.918 | 0.360 |
| INTJ | 0.173 | 2606 | 0.010 | 0.141 | 2.200 | 0.029 | 0.117 | 1.623 | 0.106 |
| INFJ | 0.364 | 4.566 | <0.001 | 0.420 | 4.495 | <0.001 | 0.317 | 2.781 | 0.006 |
| Women × PJ | −0.249 | −2.042 | 0.042 | −0.030 | −0.199 | 0.842 | 0.250 | 1.445 | 0.150 |
| Women × DJ | −0.114 | −1.128 | 0.260 | −0.139 | −1.065 | 0.288 | −0.201 | −1.459 | 0.146 |
| Women × INTJ | 0.105 | 1.162 | 0.246 | −0.037 | −0.371 | 0.711 | 0.022 | 0.227 | 0.821 |
| Women × INFJ | 0.101 | 0.842 | 0.401 | 0.020 | 0.159 | 0.874 | −0.195 | −1.334 | 0.183 |

PJ = procedural justice; DJ = distributive justice; INFJ = informational justice; INTJ = interpersonal justice; SHC = satisfaction with the head coach.

Regarding gender differences at the three time points, women showed more SHC than men at Time 0 ($p < 0.05$) and 1 ($p < 0.001$), whereas nonsignificant differences were found at Time 2 ($p = 0.192$). At Time 0 (Model 3), in the male group, PJ was a significant predictor of SHC ($\beta = -0.076$), whereas in the female group, the association was not significant ($\beta = 0.110$), and the slope difference was significant ($p < 0.05$). Furthermore, INFJ was a significant predictor at Times 0 and 1, but not at Time 2, in the female group, whereas in the male group it was a significant predictor only at Time 1. At Times 1 and 2 (Model 4 and 5), no gender differences were found.

## 4. Discussion

The purpose of this investigation was to confirm the relation between professional soccer teams' perceived coach justice and players' satisfaction with the head coach, taking into account the existing variations over time and possible gender differences. After analyzing the results, some interesting proposals can be made to advance in our knowledge of these variables, and valid applications can be extracted for coaches and sport psychologists.

The first issue was to determine how perceived justice and SHC evolve over the season. As we postulated in the first hypothesis, all the variables undergo a general decrease as the season advances, and the change between the first and last measures is significant. In spite of the fact that there are no previous studies that have analyzed these variables longitudinally, some longitudinal works have been carried out in high-performance sports with other variables related to the coach or to group processes [4,8,22]. In these studies, variables such as a more-democratic leadership, support from the coach, group cohesion, collective efficacy, or learning climate decreased as the season advanced, as in our investigation of satisfaction and perceived coaching justice. A possible explanation may be the emergence of conflicts and problems in the athlete–athlete, athlete–coach, and team–coach interactions. Initially, they may not be important but they will increase with training sessions and competitions, and, in the soccer context—and more specifically in the professional sphere examined in this work—they are very relevant due to the importance of the results and performance [9,23,29].

Regarding the second hypothesis, factors of perceived justice have been shown to positively predict players' SHC, so the second hypothesis is confirmed. Thus, of the different

factors that make up the construct, we can observe that all have predictive capacity in men or women at some point of the season. This is consistent with the results found by Nikbin et al. [19], who found that perceived coach justice stood out as an important construct to determine athletes' satisfaction in volleyball and handball teams. In this case, they combined INFJ and INTF into a single factor, which was the most important, although DJ and PJ also had significant weights.

This is very interesting because coaches' capacity to manage a soccer team depends to a large degree on how they deal with these behaviors. However, drawing on the results obtained, it is clear that not all the factors are equally important, and their importance varies as a function of moment of the season and the athletes' gender, so coaches should take these findings into account to influence the most relevant factors.

As mentioned, it was difficult to establish a hypothesis about the evolution of the different factors of justice with regard to SHC, as there are no studies that would allow analyzing prior results and because of the multitude of variables that could affect these relations, such as the players' satisfaction [19], the need for support from the coach, mastery climate, task cohesion, or identification with the team [10]. In our case, we could confirm that the moment of the season and the athletes' gender have a direct impact on the capacity of justice to predict SHC. Thus, we can observe in the results that, at the start of season, males grant more importance to aspects related to strategies and procedures to achieve adequate performance and are more satisfied with a coach who manages to develop PJ to a greater extent. However, the relational and social aspects are more important for females, as shown by the fact that INTJ predicts SHC to a greater extent. These results confirm the findings of investigations that conclude that female athletes prefer coaches with good social skills, place more emphasis on social relations, and have more empathy with the coaches than the males [1]. In contrast, males grant more importance to performance and feel more satisfied and motivated when they are better than other players [30].

In addition, the INFJ predicts the SHC in both genders throughout the season. This means that players like to be informed about why decisions are made at all times. Therefore, coaches should explain why procedures were used in a certain way to satisfy their players.

Despite this, a different evolution of the rest of the factors of justice can be observed with regard to their importance to predict SHC, taking into account the above-mentioned gender differences. Thus, males present more differences in the longitudinal analysis of the variables that predict satisfaction, as the significant variables change as a function of the moment of the season. In contrast, females seem to be more consistent over time in this analysis.

As a function of these results, we cannot state that there is a clear pattern of modification of the estimates, although, in general, in the males, the importance of the variables more closely related to the processes of training seems to decrease (i.e., PJ), and the importance of the social aspects increases (i.e., INFJ or INTJ). In contrast, the females granted a fair amount of importance to INFJ over the season, although the importance granted to the social and motivational aspects seems to decrease (INTJ), and the importance of aspects more closely related to learning and performance seems to increase (PJ).

This may be due to a different perception of the training environment and of coach behavior, as the females are more task-oriented and in general perceive a better climate in this sense, leading to minor conflicts in the group and to more enjoyment and commitment and, therefore, less decrease in satisfaction with a coach who supports them socially [24,30]. It would also be influenced by changes in the social environment of each player, although it is one of the limitations of our study that we did not measure this. Secondly, the difference at the competitive level between male and female leagues may imply that female soccer players do not value the importance of the coach for performance in the same way as the males, because all the men were professionals, whereas in female soccer, only some of the women had professional contracts. Due to this, individual and collective performance may affect group processes to a greater extent, and there may be a lesser decrease in the group's satisfaction with coach behavior and which variables are important to develop it [8,23,30].

The study carried out Kavussanu et al. [6] already confirmed that athletes' competitive level affects their motivational orientation and perception of the climate established in their near environment, so this could determine the differences found in the variables analyzed in this study.

## 5. Limitations and Future Research Directions

Among the limitations of our work, we note primarily that all the coaches of the participating teams were males, as the inclusion of female coaches is not yet very generalized. As mentioned above, this could affect the results due to the different relations established depending on whether the coach is male or female [1]. It would be interesting to know how the coach's gender could affect our results and what consequences could be drawn from these data.

Moreover, it would also be interesting to know the relations that can be established among the variables at different levels of analysis (i.e., between teams). In this study, we decided not to perform this analysis due to the small number of teams (n = 21) and due to their heterogeneity. A small number of units on a specific level in an MLM analysis could influence the statistical power of the study and could therefore also have an impact on the results [31]. A recommendation for future research is to investigate the potential differences between males and females at the team level, as well as between amateurs and professionals. Similarly, it could be very interesting to propose an intervention in which strategies to optimize justice at different moments of the season would be established in order to determine which of them is most effective and how we could modify these variables with the coaches.

Furthermore, given the importance of perceived justice to satisfaction with head coach in a professional sport environment, it can be complemented with other variables of the players' lives inside and outside of sport to be approached from a mental-health–performance perspective. In this sense, it would be useful to analyze the relationship between coach competence, coach leadership, and coach–player relations perceived by players with SHC. Furthermore, it would be interesting to consider other variables such as intrinsic motivation, according to effects observed in the decrease in stress and anxiety levels [32], as well as an increase in emotional intelligence, considered relevant for sports performance [32,33]. Furthermore, fulfilment of basic needs (feeling of autonomy, competence, and positive relations) in sport and private life, and resilience, have been identified as potential protective factors for mental health in athletes, especially during times of uncertainty (e.g., the COVID-19 pandemic and the lockdown) [34,35]. Resilient players seem to be able to promote a positive adaptation within periods of adversity [36]. Therefore, a measure of the variables that contribute to a better adaptation of athletes to their lives inside and outside of sport, as well as to their mental health, can help coaches and sports psychologists to identify the state of their athletes in order to maintain or improve performance. All this information should be taken into account for coaches and sport psychologists to individualize their response. Adaptation problems must also be detected, as well as differences between genders. In a previous study, it was found that women were more likely to report a period of overwhelming stress followed by a return to baseline wellness during the COVID-19 lockdown [37]. The authors indicated that the knowledge about your players' psychological profile may help with performing early interventions and/or perform prevention strategies. It may also be applied in coaches' behaviors, and therefore coaches should individualize the treatment to each player according to their characteristics (e.g., gender). Thus, it would be interesting if future research aimed at addressing the behavioral aspects of football players from a more global and gender-specific perspective. Furthermore, research could include measurements in the social environment of football players.

## 6. Perspective

Given the importance of the coach in the context of professional soccer, it is important to know what coach behaviors can help improve players' satisfaction, optimizing their individual and collective performance [1]. Thus, this research provides an advance in the knowledge on this topic, since it demonstrates that perceived justice is an important variable to explain players' satisfaction in the professional environment.

These findings are consistent with results found in previous studies [20,21,29], and they should be taken into account to understand coaching behavior and how it affects athletes' satisfaction. However, it should be noted that this study shows that the incidence of these variables is different in men and women, as males grant more importance to performance and females to interpersonal relations. Likewise, the effect is not the same at different moments of the season, so the coach should take this into account and know when it is preferable to focus on different aspects of justice.

Indeed, during recent years, the figure of the sport psychologist appears in semi-professional teams also. Therefore, this work should be performed as a combination between coaches, psychologist, and players. Despite the fact that the results of our study show differences by gender, psychological aspects should be worked from an individual perspective, due to interindividual differences which may influence the individual response of each player. However, the information provided in this manuscript may be used by coaches and psychologists to understand behaviors or thoughts in certain male or female players with regard the behavior of the coach. Indeed, coaches and psychologists need this information prior to starting their treatments or works. Our results suggest that coaches and psychologists should work with female players on certain social aspects such as cohesion or resilience. These may both be done with group tasks during training (i.e., coaches). For example, using tasks where the participation of all members of the group is necessary to achieve the purposes of the tasks. Meanwhile, sport psychologists can work during individual sessions with each player. For example, they may consider how they can value the behavior of the coaches or other psychology strategies. In male players, coaches may highlight the use of competitive tasks, while psychologists have the capacity to teach players how they can execute better management of the competency.

**Author Contributions:** Conceptualization, M.Á.L.-G., R.L.-M. and J.D.-G.; methodology, A.R.-M. and J.C.P.-B.; formal analysis, M.Á.L.-G. and J.D.-G.; investigation, A.R.-M. and J.C.P.-B.; data curation, R.L.-M.; writing—original draft preparation M.Á.L.-G. and J.D.-G.; writing—review and editing, A.R.-M.; R.L.-M. and J.C.P.-B.; visualization, M.Á.L.-G.; supervision, M.Á.L.-G. and J.D.-G. All authors have read and agreed to the published version of the manuscript.

**Funding:** This work was supported by the Assistance to Research Groups (GR18102) of the Junta de Extremadura (Ministry of Employment and Infrastructure), with the contribution of the European Union through the European Regional Development Funds (ERDF).

**Institutional Review Board Statement:** The study was conducted according to the guidelines of the Declaration of Helsinki, and approved by the Ethics Committee of the University of Extremadura (239/2019).

**Informed Consent Statement:** Informed consent was obtained from all subjects involved in the study.

**Data Availability Statement:** Not applicable.

**Conflicts of Interest:** The authors declare no conflict of interest.

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
