# Peer review of "The Role of Perceived Justice on Satisfaction with the Coach: Gender Differences in a Longitudinal Study"

_sustainability, doi:10.3390/su14010401_

Round 1

Reviewer 1 Report

the authors successfully addressed those issues in which I previously mentioned about. it is now ready to be published and potential readers of this manuscript would benefit from it in terms of expanding knowledge as to organizational justice and its potential influence over athletic behaviors and coaching. Appreciate your good work. 

Reviewer 2 Report

It is a satisfactory response to queries. The findings have been tied in with relevant literature and limitations are better realised.

This manuscript is a resubmission of an earlier submission. The following is a list of the peer review reports and author responses from that submission.

Round 1

Reviewer 1 Report

This study used the psychological construct of perceived justice in a longitudinal design throughout a season to determine football players' satisfaction with the coach's behavior.  Gender differences were found in factors that predict satisfaction with the coach. Notably, males were more performance-oriented, and females were more concerned with relational and motivational aspects especially interpersonal relations. It was suggested that the results could be used to promote gender and time appropriate behaviors from coaches to increase satisfaction and performance.

“… this research provides an advance in the knowledge on this topic, since it demonstrates that perceived justice is an important variable to explain players' satisfaction in professional environment”. There is longitudinal data used in the study - conclusions were made at the gender group level (not the individual level) so the relevance to sport psychologists is unclear at they generally work one on one with players. Better conclusions/ takeaway points for future research are required with regards to “valid applications … for coaches and sport psychologists”.

In the Discussion, the first hypothesis states that “variables undergo a general decrease as the season advances and the change between the first and last measures is significant”; the second hypothesis states “factors of perceived justice have been shown to positively predict players' SHC”. Then it was noted that “it was difficult to establish a hypothesis about the evolution of the different factors of justice with regard to SHC, as there are no studies that would allow analyzing prior results and because of the multitude of variables that could affect these relations”. Some ideas about these “multitude of variables” are required.

The proposal that satisfaction in a professional environment, and perceived justice – satisfaction with head coach are important factors that influence team performance, may be complemented by variables from the players’ life around and outside of sport that contribute to their mental health and wellbeing. Although, this has not been proposed yet. It is outside of the scope of the current study, but it is worthwhile to consider other studies with football players or athletes (gender focused) that report on experiences of maladjustment and resilience. A measure of adjustment issues from athletes’ life inside and outside of sport may help sport psychologists to have a more complete picture of gender variables e.g., Simons et al. (2020) found females were more likely to report a period of overwhelming stress followed by a return to baseline well-being during the COVID-19 lockdown. It is one of the “multitude of variables” but provides a pathway for coaches and sports psychologists to better understand what is going on with the players from a mental health-performance perspective.

There is a lack of discussion on whether there was an impact from changes in the social environment of football players. There could be more clarity on how coaches and mental health care systems for football players provided (or did not provide) environmental adjustment in the new-life with COVID-19 e.g., an adjustment to training and competition methods. If so, what impact did it have? What other variables may hinder or enhance perceived justice?

Author Response

Thanks for your time to improve the quality of our manuscript

Reviewer 2 Report

First of all, the title of this study captured my attention due to the word, “gender differences and coaching”. Carefully reviewed your manuscript and here are my suggetions to enhance the overall quality of your manuscript:

  1. In the introduction part, you may want to provide more indepth discussion on unique circumstance of sports and coaching prior to leading it to the discussion of percieved justice. It will help readers to be more knowledgeable about coaching and sport settings – it is not a sport specific journal so you may want to help potential readers about it.
  2. From the line 167, you mentioned CFA analysis, Cronbah’s alpha, where are those results? You should insert that figure or tables in order to deal with validity and reliability issues.
  3. From line 199, there are missing statistical information. You should clearly present those information
  4. From line 226, another missing statistical information
  5. Please check with reference list format of this quality journal and make sure your manuscript is confirming with it.

Author Response

(The authors gave the same response as above.)

Round 2

Reviewer 1 Report

I appreciate the hard work and the kind responses but there hasn't been a high- level response on the mental health aspect. Firstly, the point on better conclusion re: “valid applications … for coaches and sport psychologists” was responded to with: "Following our findings, coaches should focus on behaviors related to justice as the season advances". There hasn't been enough attention provided to sport psychologists in a take-away point/s. I see there is a referencing error for the following: "For example, it has been found that women were more likely to report a period of overwhelming stress followed by a return to baseline wellness during the COVID-19 lockdown [38]". This is not to correct article and checking it I found no possible connection. The comment was specifically referring to Adjustment Disorder in high level athletes during a COVID-19 lockdown (Simons et al, 2020) and the reference provided (Simon et al, 2021) did not refer to athletes. So I have enough concern that the paper may be rushed and not ready yet (for publishing in this journal). 

Reviewer 2 Report

Dear Authors,

I understand your diligent work to improve overall quality of your manuscript. However, I do not find much update in terms of validation process and reliability of your findings. Your manuscript does not provide much details on methodological issues especially sampling and analysis procedures. That is, one of the most critical elements of scientific research in the field of sport science. I understand you now present CFA scores but it is not enough to deal with reliability evidence of your potential findings. 

Admire your hard work but cann't let your manuscript published in this quality journal unless your clearly deal with the issues above.